# Skin-Inspired Healthcare Electronics

**DOI:** 10.3390/biomimetics10080531

**Published:** 2025-08-13

**Authors:** Saite Li, Qiaosheng Xu, Yukai Zhou, Zhengdao Chu, Lulu Li, Xidi Sun, Fengchang Huang, Fei Wang, Cai Chen, Xin Guo, Jiean Li, Wen Cheng, Lijia Pan

**Affiliations:** 1Collaborative Innovation Center of Advanced Microstructures, Jiangsu Provincial Key Laboratory of Photonic and Electronic Materials, School of Electronic Science and Engineering, Nanjing University, Nanjing 210093, China; 502023230028@smail.nju.edu.cn (S.L.); xuqiaosheng@smail.nju.edu.cn (Q.X.); 502023230057@smail.nju.edu.cn (Y.Z.); 502023230020@smail.nju.edu.cn (Z.C.); 502024230029@smail.nju.edu.cn (L.L.); xidisun@126.com (X.S.); 602024230024@smail.nju.edu.cn (F.H.); dg21230017@smail.nju.edu.cn (X.G.); jali@smail.nju.edu.cn (J.L.); 2Technology Development Center, Guangzhou Shiyuan Electronic Technology Company Limited, Guangzhou 510700, China; wangfei8398@cvte.com (F.W.); chencai@cvte.com (C.C.); 3School of Integrated Circuits, Nanjing University-Suzhou Campus, Suzhou 215163, China

**Keywords:** wearable devices, health monitoring, electronic skin, flexibility, vital signs monitoring

## Abstract

With the improvement in living standards and the aging of the population, the development of thin, light, and unobtrusive electronic skin devices is accelerating. These electronic devices combine the convenience of wearable electronics with the comfort of a skin-like fit. They are used to acquire multimodal physiological signal data from the wearer and real-time transmission of signals for vital signs monitoring, health dynamics warning, and disease prevention. These capabilities impose unique requirements on material selection, signal transmission, and data processing for such electronic devices. Firstly, this review provides a systematic introduction to nanomaterials, conductive hydrogels, and liquid metals, which are currently used in human health monitoring. Then, it introduces the solution to the contradiction between wireless data transmission and flexible electronic skin devices. Then, the latest data processing progress is briefly described. Finally, the latest research advances in electronic skin devices based on medical scenarios are presented, and their current development, challenges faced, and future opportunities in the field of vital signs monitoring are discussed.

## 1. Introduction

Vital signs are important indicators that reflect the basic life activities of the human body, such as body temperature, pulse rate, blood pressure value, and blood oxygen concentration [1,2,3]. Through real-time monitoring and timely feedback of vital signs, early health problems can be identified and detected promptly, changes in condition and prognosis can be assessed, anticipatory interventions can be implemented, and appropriate health management and preventive care can be provided [4]. For example, deaths from cardiovascular disease, such as stroke, arrhythmia, hypertension, and coronary artery disease, continue to increase. Early, rapid, and easy risk screening and intervention can significantly reduce the incidence of cardiovascular disease, and this can be achieved by monitoring and analyzing vital sign data [5,6,7].

In conventional clinical medical procedures, the detection of vital sign parameters, including heart rate, blood pressure, and blood oxygen concentration, necessitates the use of dedicated devices. As most of these devices are bulky, expensive, uncomfortable, and require specialized operation and maintenance, their widespread application is challenging in daily home care and areas with scarce medical resources [8,9]. Consequently, there is a growing demand for small, inexpensive electronic devices that do not require specialized handling [10,11]. At the same time, a single sensor is insufficient for comprehensive monitoring of human health. The prediction of numerous diseases necessitates the joint analysis of multiple vital sign signals. An important way to achieve this goal is to build wireless sensor networks containing multiple sensors [9,12]. In wireless wearable sensor networks, multiple sensor nodes are capable of simultaneously and continuously monitoring limb activity or vital sign information, transmitting these signals to a designated base station for analysis and processing through low-power wireless communication technology. This meets the requirements of high sampling frequency and continuous time synchronization, as well as precise analysis of the received signals [13,14].

The advent of flexible electronic skin (e-skin) has ushered in a new age of wearable technology, designed to enhance comfort and facilitate intuitive and comprehensive monitoring [15,16]. It is widely acknowledged that the skin is the largest organ of the human body, covering the surface of the body and in direct contact with the external environment. Accounting for 16% of the human body mass, the skin performs a number of essential functions, including protection, sensation, breathability, excretion, heat dissipation, and insulation [17,18]. E-skin, inspired by the characteristics of the skin and designed to emulate its functions, can be attached directly to or worn on the surface of the skin to perform vital sign monitoring tasks, provide feedback on the body’s health status, and assist in the intervention and treatment of medical conditions [19]. Recent research results have shown that a number of e-skin devices have achieved a high level of accuracy and reliability in vital signs monitoring and early warning of health dynamics [20,21,22,23,24]. At the same time, e-skin devices focus on user comfort, portability, integration of complex systems, and adaptability to different application scenarios [25]. In contrast to conventional bulky and rigid vital signs monitoring devices constrained by Moore’s law in terms of miniaturization limits, skin-inspired electronics are characterized by their thinness, breathability, low cost, low modulus stretchability, and ability to be worn directly on the skin, on clothing, and inside the body [26]. Such devices facilitate continuous, real-time, non-invasive, and comfortable monitoring of key vital sign parameters, providing relevant clinical indicators for the prevention, early warning, and diagnosis of a range of diseases [27].

Unfortunately, most recent reviews related to e-skin in the healthcare field have focused only on a specific type of material or a particular vital sign [28,29,30,31]. There is a lack of systematic and comprehensive discussions on the latest developments in this field. To provide readers with a comprehensive understanding of e-skin technology in healthcare, this paper offers a bottom-up overview of the field, as shown in Figure 1. We will first introduce recent advances in novel e-skin wearable materials and the key research on wireless sensing technology and data processing technology. We will then present the research on e-skin in the field of health monitoring. Finally, we will summarize this field’s development prospects and present its current challenges and opportunities.

**Figure 1 biomimetics-10-00531-f001:**
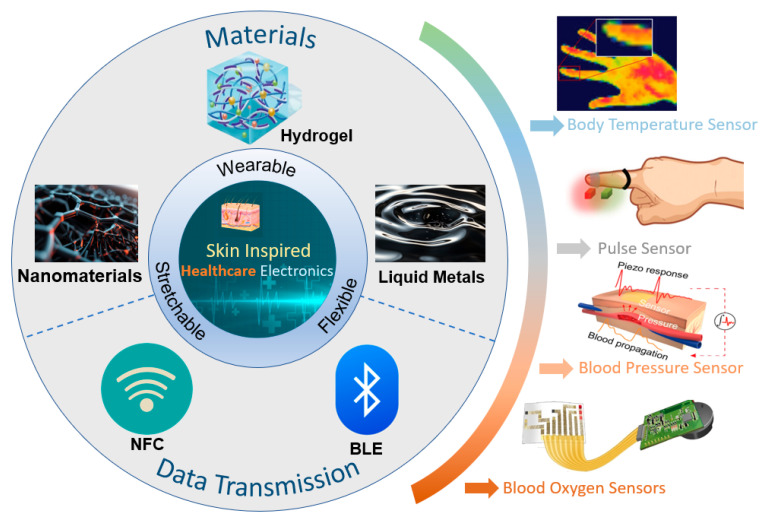
Skin-inspired healthcare electronics in terms of material selection, signal transmission, and application. Body temperature sensor. Reprinted with permission from Ref. [24] (copyright 2024 Wiley-VCH GmbH). Pulse sensor. Reprinted with permission from Ref. [32] (copyright 2023 Wiley-VCH GmbH). Blood pressure sensor. Reprinted with permission from Ref. [33] (copyright 2023 Jian Li et al.). Blood oxygen sensors. Reprinted with permission from Ref. [23] (copyright 2022 Lijia Pan, Yi Shi, Xinran Wang et al.).

## 2. Material Selection and Signaling for Skin Electronics

The foundation of bionic e-skin is the pursuit of innovative flexible wearable materials [34]. For the e-skin device to comfortably fit human skin, at least the attachment layer material of the e-skin device needs to be flexible, low-modulus, highly elastic, breathable, and thin [35]. For the device to be fully flexible, its sensor portion also needs to have high sensitivity, a wide response range, a low detection limit, and other sensing properties [36,37,38]. The development of future wearable devices must overcome the constraints of traditional rigid substrates and flat circuit board technology. At the same time, researchers must enhance the devices’ sensing performance [39]. The incorporation of highly flexible, comfortable, and stretchable conductive materials can enhance the fit of the device and reduce signal distortion caused by human skin [40]. The advancement of wireless data transmission communication technologies is also a crucial aspect of the evolution of wearable sensing systems [41]. The development of bionic skin electronics necessitates the integration of wireless data transmission technologies and high-performance novel sensing devices [42]. The objective is to enhance the monitoring platform devices’ capacity to provide real-time monitoring, timely feedback, and sustainable operation. Furthermore, it is crucial to enhance the overall system’s resilience to complex environmental disturbances, as the wearer’s daily activities can potentially impact the functionality of the electronics [43]. Various wireless communication protocols, including radio frequency identification (RFID), near-field communication (NFC), and Bluetooth Low Energy (BLE), serve as bridges for signal transmission. This high level of compatibility allows sensing systems to connect effectively with smartphones and tablets. As a result, it enables real-time monitoring, processing, analysis, warning, and response to vital signals [44,45,46].

### 2.1. Material Selection for Skin Electronics

The key components of flexible health monitoring platforms are flexible sensors, which include the substrate layer, insulating layer, and active sensing layer [47]. Additionally, the creation of new flexible conductive materials, such as electrode layers, capacitors, and transistors, is being explored for potential use in self-powered and communication systems [48,49]. In contrast to conventional materials, novel materials must exhibit superior adhesion, stretchability, and conductivity, while also being compatible with distinctive designs to fulfill the requisite specifications for diverse monitoring environments [50]. This paper will focus on three emerging categories of functional and flexible materials: nanomaterials, conductive hydrogels, and liquid metals. These materials are gaining attention due to their thinness, softness, durability, and biocompatibility.

#### 2.1.1. Nanomaterials

The term “nanomaterial” is typically used to describe materials with dimensions less than 100 nanometers, which are subject to quantum effects [51]. These materials often exhibit distinct physical and chemical properties compared to their macroscopic counterparts. They can be categorized by dimension into zero-dimensional (0D) nanomaterials, one-dimensional (1D) nanomaterials, two-dimensional (2D) nanomaterials, and three-dimensional (3D) nanomaterials [52].

Zero-dimensional nanomaterials are typically in the form of nanoparticles, including quantum dots [53] and gold nanoparticles [54], among others. However, their lack of electrical conductivity represents a significant limitation in terms of their applicability in skin electronics.

One-dimensional nanomaterials have received considerable attention and research attention, particularly silver nanowires and carbon nanotubes (CNTs), due to their excellent stretchability and tunable electrical conductivity [55,56]. Silver nanowires are mostly prepared by the liquid polyol method [57] and the self-assembly method [58], while CNTs are mostly prepared by chemical vapor deposition (CVD) [59]. Pan et al. employed silver nanowires coated with leaf veins as a pressure-sensitive layer to develop a highly sensitive microcracked flexible pressure sensor. The excellent biocompatibility, permeability, and sensitive piezoresistance of silver nanowires allow for direct connection to the human body, enabling real-time monitoring of vital signals without adverse effects (Figure 2a) [60]. However, the microcrack sensor formed by simply coating silver nanowires has been shown to substantially increase the sensitivity of the sensor. Nevertheless, it still has a monitoring blind spot for the processing of complex signals. To address this issue, the team expanded on their original design by developing a ternary material system that incorporates silver nanowires arranged in a porous, interconnected structure. This new system achieves a high sensitivity of 1167 kPa^−1^, enabling it to simultaneously monitor subtle changes in a woman’s carotid artery pulse and respiratory rate [61]. In addition to metallic nanowires, such as silver nanowires, CNTs have a wide range of applications in skin-like electronic devices. They have many advantages, such as excellent electrical conductivity, thermal conductivity, chemical stability, and energy conversion efficiency. Kim et al. dispersed CNTs onto porous, ultrathin fibrous grids, which enable conformal attachment to the dermal surface for precise, noninvasive monitoring of vital signs such as body temperature and electrophysiological signals [62]. However, poor optical transparency prevents them from being used as substrates for skin devices based on optical principles.

Two-dimensional nanostructured materials offer a very high theoretical surface area, thus resulting in novel phenomena and superior electrical properties [63,64,65]. Figure 2b illustrates graphene, the most classical two-dimensional material, renowned for its exceptional electrical conductivity, outstanding mechanical strength, and remarkable flexibility [66]. Graphene was initially prepared by mechanical exfoliation, but this method has a low yield, making it difficult to realize large-scale production. Nowadays, produced using the chemical vapor deposition method. This process involves the decomposition of carbon source gases, such as methane or acetylene, on a metal surface, resulting in a large-area, high-quality graphene film [67,68]. For example, Zheng et al. prepared a high-performance potassium-ion micro-supercapacitor by using porous activated graphene as a cathode to power a wireless pressure sensor for monitoring body movement. The supercapacitor exhibited a large operating voltage window of 3.8 V and an energy density of 34.1 mWh/cm^3^ [69]. Additionally, graphene has been further developed into graphene oxide (GO) and reduced graphene oxide (rGO). GO can be converted to rGO either by physical means or chemically [70]. Molybdenum disulphide (MoS_2_) is also a widely investigated 2D material, commonly synthesized using CVD. It exhibits remarkable semiconducting, optical, and mechanical properties [71,72]. For example, Ahn’s team prepared ultra-thin, large-scale, flexible MoS_2_-based active matrix backplane circuit-driven haptic sensors by CVD. The tactile sensor array has a pressure threshold of 1–120 kPa covering a reasonable sensing range for human skin. The low crosstalk characteristic enables high multi-touch sensitivity, allowing for the precise detection of the human body in different environments and movements [73]. MXene, a novel 2D material introduced in recent years, is characterized by enhanced mechanical properties and excellent electrical conductivity [74]. It is usually prepared by exfoliating a carbon layer from a metal nitride (e.g., Ti_3_AlC_2_) [75]. Currently, MXene has been used in different applications such as electrodes, energy reservoirs, sensors, antennas, and so on [76]. In the application of e-skin, MXene can be directly patterned using laser-induced patterning techniques to fabricate high-performance skin-like sensors [77]. Pan et al. prepared a conductive polyimide nanofiber/MXene composite aerogel piezoresistive sensor using freeze-drying and thermal imidization processes. The sensor can be used to monitor vital signs such as respiration and pulse, and has stable performance in harsh temperature environments ranging from −50 to 250 °C. This work also demonstrated that the composite aerogel of MXene has excellent oil/water separation properties, which is expected to build a more functional and integrated platform for efficient monitoring [78].

Three-dimensional nanomaterials account for more than 85% of all nanomaterials and are of interest due to the excellent selectivity, long-term stability, recyclability, and high sensitivity that can be achieved [79]. For example, the 2D material graphene produces 3D nanostructures when fabricated together with other nanomaterials such as metal oxides, metal nanoparticles, and quantum dots [80]. These graphene-based nanomaterials can be fabricated into superior biosensors such as electrochemical sensors [81], fluorescent biosensors [82], etc. Shan et al. reported an electrochemical biosensor based on graphene/gold nanoparticles/glucose oxidase/chitosan composite modified electrodes, which exhibited good current response to glucose, with a linear range from 2 to 14 mM (R = 0.999) and a detection limit of 180 μM at 0.5 V [83]. As another example, metal oxide-based 3D nanomaterials have unusual optical, electrical, and mechanical properties and are often used as alternative materials for capacitors, supercapacitors, solar cells, and clean fuel production [84]. Such 3D nanomaterials are often prepared by classical methods such as electrochemical deposition, metal–organic skeleton (MOF) oriented synthesis, hydrothermal synthesis, and electrostatic spinning [85]. A two-compartment microbial fuel cell (MFC) was prepared by Khare et al. They carbonized and activated a polymer film based on phenol-melamine precursor and prepared nickel (Ni) nanoparticle (NP)-doped carbon films with a thickness of 1 mm. The workers fabricated 3D micropillars on both sides of the carbon membrane using laser ablation and decorated carbon nanofibers (CNFs) on nitrogen-rich Ni/carbon micropillars as electrodes for MFC using chemical vapor deposition. This exposed the in situ doped electrocatalytic Ni and N within the membrane to the electrolyte and provided a 3D interface for biofilm formation. The MFCs were capable of generating a high open-circuit potential of about 0.75 V and a 2496 mW/m^2^ maximum power density, which is about ten times the power density generated by the original carbon film-based electrode [86].

#### 2.1.2. Conductive Hydrogel

Conductive hydrogels have attracted great interest as thermoelectric materials for flexible skin-like electronics due to their tunable Young’s modulus, high tensile properties, and biocompatibility [87]. As ionic conductors with excellent flexibility and versatility, their electrical conductivity is derived from the ionic motion in the hydrated polymer matrix for conduction [88]. Compared to conventional hydrogels, such as polyvinyl alcohol hydrogel and polyacrylamide hydrogel, novel conductive hydrogels have the advantages of high stretchability and high conductivity. In particular, they exhibit an elastic modulus that is highly compatible with human skin, and thus show great promise for application in high-performance skin-like electronic devices [89]. According to the different conductive substances, usually conductive hydrogels can be categorized into carbon-based, metal-ion-based, ionic-based, and conductive polymer-based. There are many methods for the preparation of conductive hydrogels, and the commonly used ones are chemical cross-linking and physical cross-linking. The chemical cross-linking method uses chemical reactions to make the polymer molecular chain cross-link, forming a network structure and introducing conductive ions [90]. The physical cross-linking method uses physical methods such as freezing, drying, etc., to introduce conductive ions so that the polymer molecular chain forms a cross-linking network [91]. Among them, the conductivity of ionic-based conductive hydrogels is usually low, while the optical transparency of carbon-based hydrogels is insufficient [92]. Furthermore, conductive polymer hydrogels are now widely used among conductive hydrogels due to their adjustable conductivity, durability under mechanical deformation, and Young’s modulus that matches that of the skin [93]. The conductivity of conductive polymers comes from the large π-conjugated system formed by alternating single and double (or triple) bonds in their main chains. When a bias voltage is applied, the directional movement of carriers through the system forms an electric current. The common methods for regulating the conductive polymer hydrogels during doping engineering are generally categorized as oxidative doping (p-type doping) and reductive doping (n-type doping) [94]. Shirakawa et al. realized the world’s first all-organic conducting polymer with conductivity comparable to that of metals by iodine doping polyacetylene for the first time in 1977 [95]. Along this line of thought, conducting polymers were then rapidly developed, including Poly(3,4-ethylenedioxythiophene)–poly(styrene sulfonate) (PEDOT–PSS), polyaniline (PANI), and polypyrrole (PPy), among many other conductive polymers commonly used today. For example, Ahmad et al. significantly enhanced the electrical conductivity of conforming conducting polymers by doping the nanomaterial MoS_2_ in PPy. The conductivity of PPy/MoS_2_ nanocomposites was about 55 times higher than that of pure PPy, reaching 8.33 s·cm^−1^. This enhancement is mainly attributed to the interaction between the lone pair of nitrogen and MoS_2_ atoms in PPy, creating additional holes in PPy [12]. The mechanical properties of conductive polymer hydrogels can be significantly enhanced through doping engineering or the introduction of unique microstructures, allowing them to better mimic and match the properties of human skin. For example, Zhang et al. constructed a highly mechanically stable conductive hydrogel based on polyacrylamide (PAM), lithium chloride, and PEDOT–PSS coated by nanofibers. This conductive hydrogel can work properly at −40 °C and can resist stretching of 0.19 MPa [96]. Introducing a second synergistic network is also a way to modulate the mechanical properties of conductive polymer hydrogels. The synergistic network can compensate for the hardness and fragility of natural conductive polymers, allowing them to withstand greater deformation. Tan et al. developed a self-adhesive conductive polymer composite by blending a mixture of a PEDOT–PSS composite and an elastomeric polymer network (chemically cross-linked PVA network with glutaraldehyde) (Figure 2c). The supramolecular solvents (citric acid and cyclodextrin) can form hydrogen bonds with the PSSH units and interact electrostatically with the positively charged PEDOT. This inhibits the aggregation of PEDOT chains and thus improves the mechanical flexibility of the conductive polymers. Meanwhile, PVA is constructed from PEDOT–PSS composites, which enables the material to obtain large and reversible stretchability. The modulation of the synergistic network allows the material to have a tunable and low modulus of 56.1–401.9 kPa and a high stretchability of 700% [97].

However, the biggest problem with conductive hydrogels in current applications is the lack of stability [98]. For instance, when moisture is lost, there is a significant drop in performance. As a result, skin devices based on conductive hydrogel are often susceptible to temperature, humidity, light, and other factors, resulting in a limited environment for their use [99]. Currently, researchers are trying to achieve longer-term stability with methods such as encapsulation, finding new materials, and new mechanisms [100]. Among these, Yuk et al. used thin elastomer films robustly bonded to hydrogels to form anti-dehydration hydrogel–elastomer hybrids. They conducted dehydration experiments comparing hydrogel discs with and without elastomer films under environmental conditions (24 °C and 50% humidity). After 48 h, compared to hydrogels without elastomer films, which lost weight to nearly their original water content (∼85 wt.%), the anti-dehydration hydrogel–elastomer hybrids showed no significant weight change over 48 h [101]. Bai et al. introduced highly hydratable salts into polyacrylamide hydrogels to enhance their water retention capacity. Specifically, polyacrylamide hydrogels containing a high concentration of lithium chloride retained over 70% of their initial moisture content even under conditions of 10% relative humidity [102]. Further breakthroughs are anticipated.

#### 2.1.3. Liquid Metals

Liquid metals are characterized by superb self-healing ability, high electrical conductivity, superb stretchability, and conformability due to their dual properties of both liquid and metal [103]. In recent years, with the emergence of gallium (Ga) and its alloy liquid metals, which are microtoxic and well biocompatible, liquid metals have developed a variety of applications for wearable electronic devices such as stretchable electrode materials, wearable antennas, skin-applied heaters, flexible sensing layers, and flexible neural synaptic devices [104]. However, due to the fluidity and difficult-to-regulate surface tension of liquid metals, how to process liquid metals into desired patterns is an urgent problem to be solved [105]. A common approach is to prefabricate microfluidic channels using elastomers such as PDMS, SEBS, etc., and inject liquid metal into their internal microchannels to pattern them. Gao et al. demonstrated a microfluidic haptic diaphragm pressure sensor with embedded liquid metal microchannels. The pressure sensor with a flexible wristband monitors the pulse in real time and is able to resolve pressure changes of less than 50 Pa with a detection limit of less than 100 Pa and a response time of 90 ms. The embedded equivalent Wheatstone bridge circuit fully utilizes tangential and radial strain fields to achieve a highly sensitive output voltage variation of 0.0835 kPa^−1^ [106]. However, the usual method of injecting liquid metals lacks sufficient flexibility and convenience and can be limited in large-scale commercial production and use. Liquid metals, especially Ga-based liquid metals, form self-limiting gallium oxide layers on their surfaces. Removal of the oxide layer is required to achieve control of mobility and to utilize its properties for the preparation of special structures. The surface oxide layer can be effectively removed by subjecting Ga-based liquid metals to acidic or alkaline environments. Sun et al. used an alkaline solution method to prepare liquid metal electronic devices with leaf vein structures. They used a high concentration NaOH solution to remove the surface-restricting oxide layer from the liquid metal, making it compatible with photolithography. This process enabled the design of stretchable electronic circuits on the human epidermis. The electrical device can function not only as a stretchable electrode, but also as a heater and a strain sensor, realizing the stable operation of the electronic device attached to the human epidermis during human movement [107]. Through high-precision pneumatic 3D printing technology, Park’s team realized electrophysiological analysis of retinal ganglion cells in early to mid-stage retinal organoids by directly printing liquid metal into three-dimensional columnar electrodes (Figure 2d). Since the Young’s modulus of the liquid metal electrodes matches that of human organs, the synaptic structure of the liquid metal electrodes is less invasive to the soft and delicate organs of the retina, allowing long-term monitoring of neural activity [108].

In conclusion, the application of liquid metals in e-skin devices is still in its infancy, but its broad future prospects can already be foreseen. In particular, Ga-based liquid metals, which can act as circuits, electronic components (resistors, diodes, transistors, and neural synaptic devices, etc.), wearable antennas, and other kinds of skin-applied devices, have been touted as the future of e-skin [109,110].

**Figure 2 biomimetics-10-00531-f002:**
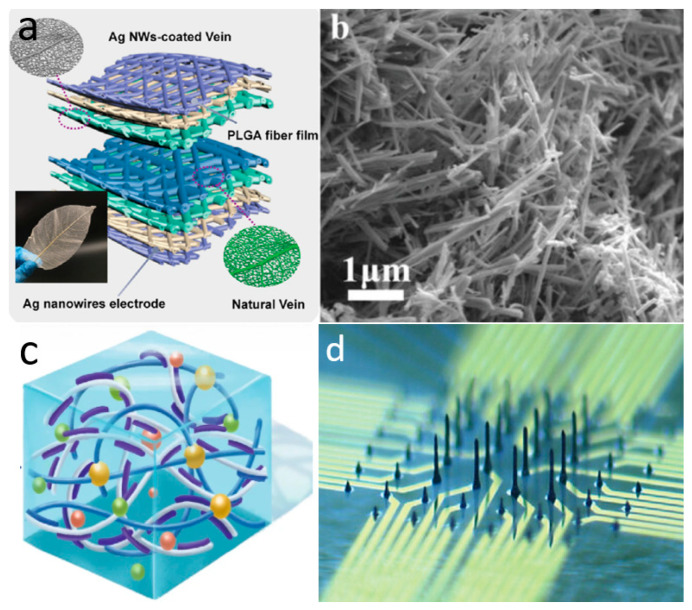
Material selection for novel skin electronics. (**a**) Microcracked flexible pressure sensor prepared using 1D nanomaterial silver nanowires. Reprinted with permission from Ref. [60] (copyright 2021 Wiley-VCH GmbH). (**b**) Two-dimensional nanomaterial porous graphene. Reprinted with permission from Ref. [69] (copyright 2021 Wiley-VCH GmbH). (**c**) Highly mechanically stable conductive hydrogel based on PAM, lithium chloride, and PEDOT–PSS covered by nanofibers. Reprinted with permission from Ref. [111] (copyright 2022 Wiley Periodicals LLC). (**d**) Liquid metal columnar electrodes made by 3D printing technology. Reprinted with permission from Ref. [108] (copyright 2024 Sanghoon Lee, Won Gi Chung, Han Jeong et al.).

### 2.2. Wireless Data Transmission for Bionic Skin Devices

Users receive vital indications from e-skin and e-skin-like devices that are continuously improving in accuracy and comprehensiveness [112]. Simultaneously, the need for quick, differentiated monitoring, diagnosis, and treatment of various user groups is increasing [113]. This raises the bar for wearable device monitoring and communication capabilities. Wireless connection technology can improve the monitoring system’s mobility and comfort, and provide real-time data synchronization and feedback. It also facilitates more intelligent interactions and personalized services, secures data signals, and safeguards user privacy [114]. The ultimate objective is to quickly and efficiently wirelessly send dynamic physiological signals to smart devices, enabling qualified medical professionals can diagnose patients. Emerging wireless communication technologies like NFC and BLE are currently being used in skin-like devices. These technologies are safe and practical mainstream wireless data transmission technologies that offer concepts for dynamic response, real-time device monitoring, and instant feedback.

#### 2.2.1. Near-Field Communication

The realization of skin-like electronics for wear and application inevitably requires reference to the Internet of Things (IoT). IoT is an information carrier based on the Internet, traditional networks, etc. It allows all common physical objects that can be independently addressed to form an interconnected network. Information interaction between people and things, and between things and things, is the core of IoT [115]. RFID is a traditional, important, and common application solution for IoT systems. RFID is not only a communication that enables two-way communication in a state of contact between two objects, but also has a unique identifier and can be equipped with various sensing functions. RFID was initially used for anti-counterfeiting and anti-theft purposes [116]. It includes different operating frequency bands such as low frequency, high frequency, and ultra-high frequency. Low-frequency RFID has been developed in commercial applications for many years due to its strong anti-jamming properties; however, the efficiency of data synchronization and transmission is relatively low. Ultra-high frequency RFID has a very wide transmission range, but the exclusive reader is priced at more than USD 1000, and by the environmental impact of easy loss and detuning [117]. NFC is also regarded as a subset of RFID since it is founded on RFID technology, together with advancements in wireless interconnection technology. Its primary function is the utilization of a single 13.56 MHz band for short-range devices to facilitate quick communication [118]. NFC technology can realize two-way interaction between electronic devices, with high security and confidentiality, making up for the lack of traditional RFID areas. NFC tags and collectors can communicate with each other peer-to-peer (P2P), which is not possible with other RFID technologies. In addition, NFC also supports battery-free mode operation, which allows for the synchronization of power transmission and wireless data transmission. Therefore, NFC communication or direct NFC sensors are a promising tool for vital sign signal monitoring in skin-like wearable device applications.

The ultimate need for flexible electronic platforms that can be used to integrate skin-like electronics and wireless communication technologies is the real-time synchronized transmission of relevant monitoring information data to the user or back-office physician. For instance, sensors used to monitor vital signs in surgical patients, neonates, and particularly preterm infants need to be smaller in order to be worn inconspicuously and have low power consumption for continuous, safe monitoring [119]. NFC chips with low power consumption are an important development in terms of energy harvesting capabilities. In addition to sending radio frequency (RF) signals, the NFC receiving antenna can also be used to recover electromagnetic energy and power the entire receiving device. Signal transmission systems in wearable applications on human skin or other portable modes are usually passive. In this condition, the NFC sensor collects energy from the RF signaling in a passive mode and couples it with the magnetic induction of an active device to obtain a wireless energy supply. By performing wireless sensing interaction tasks and transmitting data with active devices (such as smartphones) in a read–write fashion, the NFC module overcomes the drawbacks of conventional battery-worn sensing systems and cable connection modes. NFC sensor tags include semi-passive mode and passive mode (no battery). In semi-passive mode, the NFC module can realize continuous and uninterrupted monitoring of human vital signs and independently run the collection and transmission functions. The semi-passive mode’s life usually ends when the battery is fully depleted and the passive mode is activated, where the monitoring data is stored in non-volatile memory. In passive mode, the NFC tag collects energy from multiple sources such as sound, light, heat, and vibration. Both passive or semi-passive mode NFC tags can obtain energy from external devices from the magnetic field generated by the reader [118].

The importance of the NFC antenna is not only reflected in the wireless communication and energy collection aspects mentioned above, but its design is also reflected in the aspects of energy conversion efficiency and wearing comfort. First, since the operating frequency and the communication distance are essentially set, the coil size of the NFC tag antenna defines the quality factor (Q factor) and the data reading range. These factors are crucial for the real-time monitoring system. The coupling factor K for the same two loop antennas at an axial distance X depends on their respective radii, r_1_ and r_2_. The best case is when the radii of the two loops are approximately the same (r_1_ ≈ r_2_) [120]. And when the NFC tag is close to a smart reading device, such as a cell phone, its antenna usually reduces the wireless power transmission capability due to detuning. Q can be obtained from Q = Im(Z)/Re(Z), where an increase in the distance between the receiving and transmitting devices results in a decrease in the antenna’s Q-factor and input voltage. Also, the resonant frequency decreases due to the change in capacitance. Therefore, in the best case, the maximum distance between the tag and the reader to acquire the tag is 2 cm. Another very important factor is the flexible antenna fabrication technology that conforms to the wearability of human skin [121]. Currently, the mainstream preparation process is still on the flexible substrate through the technology of photolithography to achieve the flexibility of the coil. In order to wirelessly charge a stretchy battery, Xu et al. used photolithography and etching to create a stretchable wireless coil with a self-similar snake pattern on PI following the sputter deposition of a conformal Cu (600 nm) layer on PI. The power of the primary coil is 187 mW, and the efficiency of the secondary coil is up to 17.2% at a distance of 1 mm between the primary and secondary coils [122]. Although photolithography can produce precise wiring and high-quality preparation, its use in skin-like electronics is limited by its high cost, incompatibility with flexible substrates, and complicated preparation flow and process. Printing technology is now an emerging antenna preparation technique. Copper inks have been used to fabricate NFC antennas by screen printing [123]. A printing plate is used to prepare the desired pattern from copper ink under pressure, which can be transferred to flexible substrates such as paper. Its inductance is similar to that of a silver paste antenna, with a return loss of only −16.8 dB. One of the most promising methods for preparing flexible antennas is inkjet printing, which can fully overcome the drawbacks of photolithography or printing and enable the manufacturing of electrical devices on a wide scale [124]. Furthermore, this technique can regulate the printed coil’s thickness to successfully address the low Q issues of the other two approaches: if the antenna’s thickness is less than 40 μm, the theoretical Q can reach 90 [125]. Wang et al. prepared flexible metal antennas with high adhesion and low resistivity by inkjet printing combined with surface modification and chemical deposition on RFID tags at low cost [126]. The thickness of the antenna is adjusted by inkjet printing, and when the thickness of the copper layer is less than 1.1 μm, the reflection is reduced by more than 15 dB. When its thickness is higher than this value, the return loss of the antenna decreases. This conclusion suggests that the antenna can exhibit the best working performance when the copper layer thickness is 1.1 μm. However, how to effectively improve the conductivity of flexible circuits prepared by inkjet printing is still a challenge to be faced.

#### 2.2.2. Bluetooth Low Energy

Bluetooth technology has a longer data transmission range than NFC, and is a wireless signal transmission technology that supports devices to communicate over short distances (3–100m) [127]. It enables wireless data transmission and exchange between devices including smartphones, laptops, wireless headphones and numerous IoT smart devices. BLE emerged after the release of the Bluetooth 4.0 specification in 2010 [128]. Our primary recommendation is to protect personal information from disclosure, given the growth in personal healthcare and the requirement for data privacy and security in smart analytics programs related to care and monitoring. Liu et al. proposed an extended design of BLE, which safeguards the security of the patient’s data by modifying the encryption mechanism and the BLE pairing method (symmetric encryption algorithm and asymmetric encryption algorithm) [129]. By integrating a sensing module and a BLE module into an energy harvesting device to enable wireless sensing, Kwon et al. designed a remote automated system for monitoring and evaluating dermatological conditions and drug efficacy capabilities [130]. Powered by a lithium-based polymer battery, the BLE module is integrated into the chip to form a control system for remote wireless operation, allowing rapid and accurate assessment of skin hydration and skin barrier function. The BLE module enables simple and fast access to assessment metrics and data from a smartphone, providing a possible line of frequent and continuous use in the home environment. Neonatal vital signs monitoring also requires very precise signal acquisition and a foreign-body-free monitoring platform. A foreign-body-free monitoring apparatus and extremely accurate signal collection are also necessary for monitoring neonatal vital signs. To ensure standards for neonatal and pediatric intensive care, the BLE communication module enables accurate and important signals to be transmitted continuously and at high speed within a space with a diameter of up to approximately 10 m. Rogers’ team has proposed wearable electronic devices with integrated BLE wireless data transmission for continuous monitoring of maternal, fetal, and neonatal movements and vital signs [131]. They prepared a highly integrated sensing platform that can provide real-time, continuous vital signs monitoring for both mother and fetus by utilizing flexible electronics and the high compatibility of the BLE communication system. These include three sensors in the chest, limbs, and abdomen that can be operated synchronously in real time for cuff-less blood pressure monitoring, electrocardiogram-derived uterine monitoring, and tracking body position. All three sensors are connected to a central mobile device that acts as a base station for BLE communications and real-time data reception. The data is then processed algorithmically to analyze vital sign parameters. This also avoids the risks of conventional systems that lead to medically induced skin damage, complicate clinical care, and impede skin-to-skin contact between parent and child. Through the BLE communication protocol, a wireless transmission scheme is utilized between the main sensor and other sensors to synchronize local time and reduce data latency. Every sensor continuously gathers signal responses and wirelessly sends the information to a nearby device for processing and analysis. This device processes the motion data that each sensor records as well as vital sign data, including temperature, respiratory, and cardiopulmonary data that the chest and abdomen sensors record.

Compared to traditional Bluetooth, BLE offers lower power consumption and faster transmission rates. Currently, BLE communication technology is relatively mature and has been widely adopted in commercial applications. Although some BLE chips and modules have achieved compatibility with flexible substrate materials and technologies, issues such as their large physical size and reliance on external power sources remain unresolved [132].

NFC excels in “stability and low power consumption,” making it suitable for short-range, small-data instant interactions; Bluetooth leverages its advantages of “long range and high speed,” making it suitable for scenarios requiring continuous connectivity or large-data transmission, but it must compromise on stability and power consumption. Health monitoring e-skin should select an appropriate wireless data transmission method based on its specific medical application scenario.

### 2.3. Data Processing and Analysis for Bionic Skin Devices

As the sensitivity, sampling rate, and number of vital sign sensors increase, the amount of data generated inevitably grows exponentially. While this change improves the accuracy of e-skin devices and disease diagnosis, it also raises some issues. Higher sensitivity and sampling rates cause instruments to faithfully record various types of noise alongside valid data, resulting in unnecessary data redundancy. This increases the burden on data transmission and computation in later stages and leads to additional energy consumption [133]. Therefore, it is necessary to perform appropriate preprocessing of signals to filter out noise, reduce computational burden, and improve data and energy efficiency.

Frequency-domain filtering is one of the traditional noise reduction techniques. This method involves adding bandpass filters and notch filters to the circuit to remove unwanted frequency-domain signals, thereby achieving the objectives of separating the target signal and eliminating motion artifacts [134]. It is still widely used in instruments for detecting electrocardiogram [135], electroencephalogram [136], and electromyogram signals [137]. Meanwhile, emerging machine learning and deep learning technologies have also begun to be used for data preprocessing. In addition to filtering out noise, machine learning can integrate data from multiple sensors and compress the obtained data. For example, Hong et al. used machine learning to exclude pre-stimulation and tactile stimulation information, focusing solely on post-stimulation synaptic strength values. Ultimately, even with the reduced dataset accounting for only 7.5% of the total signal, the average accuracy rate still reached 96.56% [138]. However, executing machine learning tasks requires stringent computational resources. Therefore, specialized hardware solutions with better performance and energy efficiency have become a more viable option compared to traditional computing architectures [139].

However, although signal preprocessing is very useful, signal distortion is still almost unavoidable. At the same time, high-precision machine learning operations mean higher energy consumption and latency. Therefore, it is still necessary to reduce noise in advance in terms of device structure and measurement principles.

Increased data volume also means increased computational complexity. Therefore, advanced data analysis methods are equally necessary. Currently, e-skin devices for cardiovascular monitoring, fall detection, and disease prediction often utilize machine learning for data processing due to their high computational complexity. These wearable devices typically transmit data via Bluetooth to microcontrollers, smartphones, or computers for processing. These devices enable intensive machine learning training tasks, allowing users to select appropriate architectures based on specific requirements [140]. Utsha et al. utilized a customized wearable electrocardiogram (ECG) data acquisition system to transmit ECG data in real time to a smartphone. They trained various models using the MIT-BIH Arrhythmia Dataset and ultimately found that the Long Short-Term Memory (LSTM) model performed the best, achieving an accuracy rate of 95.94% in detecting and classifying heart diseases [141]. Pankaj et al. employed an automatic feature extraction deep learning method to predict heart rate in real time using photoplethysmography (PPG) sensor signals. They trained and tested deep convolutional neural network (CNN) models using two publicly available PPG datasets and further evaluated the models using internally collected PPG signals. The proposed framework achieved a correlation coefficient of 0.997 in testing, demonstrating its capability to estimate heart rate using PPG signals [142].

Although neural networks are increasingly widely used, there has been little discussion of the principles behind model selection. There is still a lack of comparative analysis of different models in various medical settings. In addition, how to perform machine learning calculations under conditions of low power consumption and limited storage and performance is another issue that researchers need to continue to consider.

## 3. Applications of Skin Electronics

The emergence of bionic skin electronics with multiple functions that can be worn for extended periods of time has been made possible by the ongoing advancements in flexible electronics technology, the development of more pliable, breathable, and comfortable materials with superior electronic properties, and the realization of wireless charging and transmission technology [143]. Bionic skin electronics have the ability to continuously test different human body indicators in real time over a longer period of time than rigid, large-scale devices that are more likely to be placed in hospitals for brief periods of time for precise testing. They are also more dedicated to being conveniently used in everyday life scenarios [144]. This kind of continuous, real-time, non-sensory measurement of different human body indicators offers data support for ongoing chronic disease monitoring, early illness detection, and more opportunities for the expansion of therapeutic approaches in both time and space [145]. This section will focus on describing the application of bionic skin electronics in monitoring health status, such as body temperature, pulse rate, blood pressure, and blood oxygen. It is foreseen that future health monitoring systems will be integrated, multifunctional, and miniaturized.

### 3.1. Body Temperature Sensor

Body temperature is one of the most important indicators of the human body’s health status. Measurement of both body and skin temperature is essential for medical decisions in clinical settings, particularly in conditions such as sleep disorders, infection, inflammation, and childbirth [146]. For patients with conditions such as stroke, continuous monitoring of body temperature is crucial [147].

Currently, the vast majority of clinical temperature monitoring is only discontinuous measurements in isolated locations such as the oral cavity, axilla, temporal artery, or rectum [148]. On the one hand, such measurements need to be performed manually by clinicians or caregivers every time, which is not only labor-intensive, but also has a large window period and cannot continuously respond to the patient’s temperature data in real time; on the other hand, these measurement devices are usually rigid, which is not only not guaranteed in terms of comfort, but also inconvenient for groups requiring special care (e.g., infants). Therefore, there is an urgent need to develop wireless flexible skin temperature sensors that can continuously monitor body temperature.

A typical skin-like temperature sensor usually consists of two parts: a sensitive layer that converts the temperature into an electrical signal and a transmission layer for data processing and transmission [149]. An encapsulating layer with strong flexibility, biocompatibility, and excellent heat conductivity will be added if the sensitive layer’s material cannot come into direct contact with the skin due to comfort concerns, potential toxicity, or reactions with perspiration or air.

Different sensitive materials can be used to construct various temperature sensors, such as thermistor temperature sensors [150], pyroelectric temperature sensors [151], or potential sensors [152]. The most commonly chosen temperature sensors are thermistor temperature sensors, pyroelectric temperature sensors, and potential sensors. Among them, thermistor sensors have become the preferred solution for converting temperature into electrical signals due to their ease of use, low cost, and simple structure. Currently, research on various wearable thermistor sensors based on low-dimensional carbon materials, conductive polymers, two-dimensional semiconductor materials, and their composites has blossomed. Thermistor sensors are capable of obtaining the actual temperature by measuring the resistance value and performing simple calculations. Hydrogel temperature sensors, for example, can be converted from resistance difference to temperature by the following formula:(1)$$\frac{\Delta R}{R_{0}}=m\cdot exp(-\frac{\varepsilon_{a}}{\kappa_{B}T})+D$$
where R_0_ is the impedance reference value, Δ*R* is the difference between R_0_ and the impedance at the real-time temperature, *m* corresponds to the material properties, $\varepsilon_{a}$ is the activation energy, $\kappa_{B}\;$ is the Boltzmann’s constant, and *D* is the fixed parameter of the individual device. Since the resistance value is a characterization of the temperature, the construction and materials of the resistors usually need to be specially designed to exclude the effect of factors other than temperature (e.g., stretching and twisting) on the resistance value.

Due to the maturity of the thermistor field, innovations in wearable skin temperature sensor research at this stage are either focused on flexibility, resolution, and comfort, or on wireless transmission and multimodality. Zhang et al. have innovated in materials and synthesized a wearable ionic temperature sensor array based on ionically crosslinked polyacrylamide-sodium alginate (PAAm-SA) hydrogel, which has a robust, responsive, and temperature-resolved hydrogel temperature-sensitive layer that is not affected by deformation (Figure 3a). The sensitive layer of the device, PAAm-SA hydrogel, was synthesized from polyacrylamide and sodium alginate by a two-step cross-linking method. It combines the inherent ionic conductivity of the hydrogel with the skin-like mechanical properties brought about by the introduction of PAAm. The material is essentially unaffected by deformation since the change in conductivity is caused by the change in mobility of ions in the hydrogel with temperature. This allows temperature measurements to remain reliable when the device is squeezed and twisted. Benefiting from a wide sensing range (22–100 °C) and a high response time (2.02 s for a temperature difference of 40 °C), the device monitors human respiratory behavior through temperature differences (0.5 °C due to breathing). Based on this temperature sensor, the researchers fabricated sensor arrays to range the human body temperature with an optimal resolution of 0.15 mm^−1^ [24]. At the forefront of the industry, Rogers et al. have invented a non-invasive, non-sensory, millimeter-scale sensor that can be placed on the contact surface between the residual limb and the prosthetic arm of a limb-disabled person to detect temperature and pressure at the skin-prosthetic interface. The sensors are placed at multiple key nodes at the skin-prosthetic interface, providing continuous wireless, continuous monitoring of temperature and pressure. A commendable point is that temperature, in addition to being used as an indicator of the human body, is used in this device to eliminate the effect of temperature on the pressure sensor accordingly. The final pressure is determined by the following equation:(2)$$P=n\;\cdot \;(\mathrm{ADC}\;-\;m\;\cdot \;(s\;\cdot \;{\mathrm{ADC}}_{t}\;+\;c_{1})\;-\;c_{2})$$
where ADC is the pressure value from the ADC channel, ADC_t_ is the temperature value from the ADC channel, and n, m, s, *c*_1_, and *c*_2_ are constants obtained from the calibration process. Every sensor has a wireless module attached to the exterior of the prosthetic limb that powers it wirelessly, sends the data to a different portable electronic device (such as a tablet or smartphone), and shows the results instantly. During clinical measurements of two prosthesis users, the temperature sensor accurately reflected changes in temperature while the test subjects were sitting, standing, and walking. The authors say the device can consistently and accurately capture temperature and pressure levels at the interface without irritating or damaging the patient’s skin [153]. Inspired by the fusion mechanism of vertebrate skeletal muscle and skin, Bai et al. have invented a soft robot that can be used as an electronic implant (Figure 3b). It can be used to detect temperature, pressure, and strain in various parts of the human body, and has functions such as applying stimuli and administering drugs. The soft robot is composed of an e-skin and artificial muscles, integrated with multifunctional sensors and drive systems, and can be used as a wireless biocompatible platform for adaptive motion. Researchers have incorporated temperature-sensitive materials into a polymer matrix through a solution-based in situ approach, resulting in a temperature sensor with high spatial and temporal resolution. The soft robot can creep at 32 mN after implantation and perform temperature measurements after entering the target area. The temperature coefficient of resistance (TCR) is as high as >0.5%/°C. The temperature results will be transmitted to an external terminal via wireless communication [154].

As of right now, temperature measurement technology is well established. The limiting factor for the development of wearable skin temperature sensors is instead the technology for communicating over long distances in complex environments, as well as the issues of cost, comfort, and reusability of these modules, which are issues that all skin-inspired electronics need to face.

**Figure 3 biomimetics-10-00531-f003:**
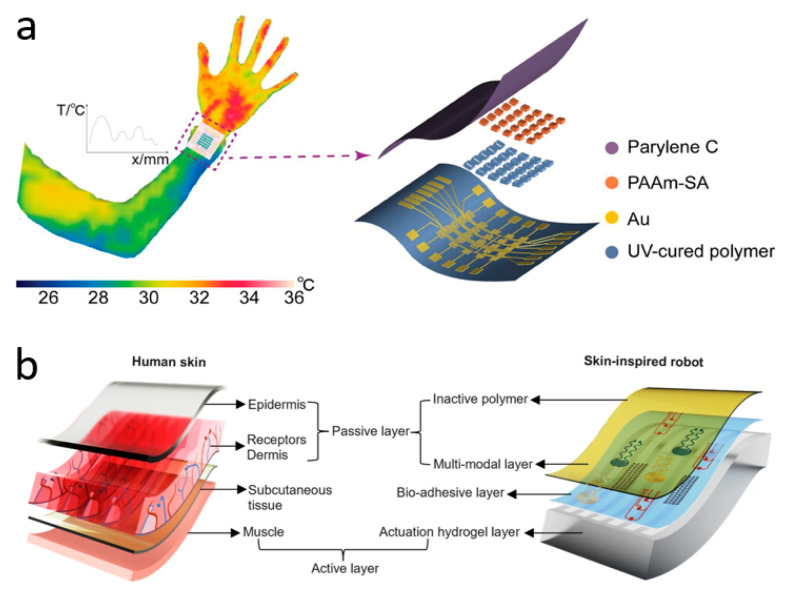
Temperature sensors. (**a**) Wearable ionic temperature sensor arrays with hydrogels that can detect temperature distribution at the wrist. Reprinted with permission from Ref. [24] (copyright 2024 Wiley-VCH GmbH). (**b**) Skin-inspired soft robots that can be wirelessly controlled to enter the human body to measure temperature, etc. Reprinted with permission from Ref. [154] (copyright 2024 Lin Zhang et al.).

### 3.2. Pulse Sensor

A pulse wave is also an important physiological signal of the human body, reflecting a variety of human health information [155]. Monitoring the pulse wave is an important way to monitor human health. For example, pulse waves can reflect states such as medication, exercise, and sleep. By monitoring subtle pulse wave changes, diseases such as aging blood vessels and heart rate disorders can also be diagnosed before symptoms appear [156]. Traditional Chinese medicine attaches great importance to judging the state of the human body based on the pulse wave, but there are often cases of misjudgment due to the lack of conditions for continuous monitoring of the pulse and the great uncertainty of the empirical diagnosis method [157]. To better utilize the pulse wave to determine health status, we need continuous pulse monitoring data for quantitative analysis [158]. Therefore, it is important to explore how to reliably monitor the pulse continuously.

Currently, the most widely used clinical pulse measurement method is PPG, a non-invasive method of optically detecting blood circulation that reflects cardiovascular status in real time [32]. As shown in Figure 4a, the beating of the heart causes periodic contraction and diastole of blood vessels throughout the body, and causes periodic changes in the volume of blood vessels. When using an LED light source to irradiate human blood vessels and tissues, the absorbed and reflected light will be affected by the vascular volume. By using a receiver to capture the reflected and absorbed attenuated light, the changes in vascular volume caused by pulse pressure can be determined, and the pulse wave can be extracted. Figure 4b shows the main components of the PPG signal: the AC signal caused by flowing blood cells, which responds to parameters such as pulse rate, heart rate, etc.; and the DC signal caused by a light source affixed to the skin, which penetrates tissues such as fat, muscle, and bone in addition to blood. The method typically uses infrared and red light, as these wavelengths are more penetrating. Recently, some researchers have also tried to use green light as a light source to obtain a higher signal-to-noise ratio because oxyhemoglobin and deoxyhemoglobin absorb green light better. Depending on the relative position of the emitter and receiver, sensors can be categorized as reflective or transmissive. Sensors based on the transmitted light principle must be placed in thinner areas of human tissues, like the tongue, earlobes, and fingertips, due to the extreme attenuation of light propagation in human tissues. In contrast, sensors based on the reflected light principle can be placed in a greater variety of places, like the wrists, necks, and so on. Thanks to the simpler structure of the PPG system and the fact that the required LED light source and sensors can now be miniaturized to a level that does not affect the overall bendability of the device, the PPG method is well suited to the flexible and comfortable requirements of electronic skin. For the purpose of monitoring human pulse and limb swelling level in daily life, Wu et al. proposed a PPG-based skin-like photovoltaic system that can meet the dual requirements of mechanical stability and wearing comfort. A flexible encapsulated inorganic LED as an emitter of red light at 635 nm wavelength was placed on the fingertip for receiving the reflected light from blood vessels using a flexible organic-inorganic hybridized chalcogenide photodetector. The photodetector has a maximum power density of 64.4 mW/cm^2^ at a current of 8.33 µA and a switching ratio of 3.2 × 10^3^. By analyzing the signal feedback from the photodetector, the researchers were able to analyze the frequency of the user’s pulse wave and the degree of swelling of the limb. The flexible substrate used for the platform ensures overall flexibility and shape adaptation when the device is attached to the finger. The wrinkled serpentine-shaped electrical interconnect structure designed on the substrate also allows the platform to adapt to the ductility of the skin [32].

Although PPG technology is very sophisticated and convenient, it is susceptible to skin color, perspiration, light intensity variations, and the distance between the device and the artery. In addition to these limitations, PPG requires an external power source, which makes the entire measurement system bulky and difficult to wear for extended periods of time for monitoring [159]. To address these issues, wearable pressure sensors for pulse wave monitoring were created and have grown by leaps and bounds. Due to their exceptional wearability and durability, pressure-sensitive sensors are extensively employed in a wide range of continuous, non-invasive wearable devices. They come with simple structures, great chemical stability, light weight, low cost, and the ability to be worn for extended periods of time. Wearable pulse pressure sensors can be tightly attached to human skin for a long period of time, and utilize mechanisms such as piezoelectricity, optics, magnetoelasticity, friction electricity, or capacitance to continuously and accurately convert pulse into electrical signals for subsequent diagnosis and analysis (Figure 5a) [160]. Aiming to be senseless, ultra-thin, ultra-light, and breathable, Zhang et al. constructed a unique multifunctional electrospun micro-pyramid array (EMPA) by the electrostatic spinning self-assembly technique (Figure 5b). The array can be used as a hybrid piezoelectric-friction electric pressure sensor to monitor ultra-weak fingertip pulse for health diagnosis. With a thickness of less than 50 μm, a mass of only 1.1 milligrams per square centimeter, and exceptional breathability, the EMPA film is a great option for achieving the concept of invisible skin. In total, 95% of participants said that the EMPA device had no effect on their regular lives or work. EMPA-based radiant cooling fabrics also have high visible-near infrared reflectivity and high mid-infrared emissivity, which can reduce skin temperature and provide long-lasting comfort in light environments. Furthermore, EMPA membranes also have excellent electrical properties such as high sensitivity, ultra-low detection limit, and biomechanical energy harvesting capability. These qualities allow the EMPA-based piezoelectric capacitance sensor device to continuously monitor a driver’s ultrasubtle fingertip pulse over long periods of time in complex driving environments. It records more details of physiological indicators than commercial photovoltaic volumetric tracers, while avoiding the problems of ischemic pressure necrosis and mechanical damage that can occur with them. In clinical trials involving e-sports players who frequently press their fingertips, the fingertip pulse sensor remained stable after 550 clicks and 20 heavy presses of approximately 20 N. This proves the mechanical stability of the device. Furthermore, EMPA nanogenerators with high triboelectric and piezoelectric outputs can even achieve energy harvesting [161].

Wide-scale application of pulse-monitoring devices may be an opportunity to free people from reliance on centralized hospitals for health monitoring anytime, anywhere, and to enable personalized, specialized medicine [162]. However, eliminating the interference of human movement on pulse measurement, ensuring long-term comfort and robustness while fitting the body, and accurately capturing valid pulse information are all critical challenges that researchers must continuously address [163].

### 3.3. Blood Pressure and Oxygen Sensors

Continuous monitoring of blood pressure and blood oxygen under non-clinical conditions is important to keep track of a patient’s health status. The initiative enables early prediction or monitoring of cardiovascular diseases such as nocturnal hypertension, heart failure, and peripheral arterial disease in a daily setting [164]. More importantly, it allows for the correlation of blood pressure and oxygen trends with an individual’s daily habits and lifestyle, enabling healthcare professionals to analyze the underlying causes and provide recommendations, interventions, and treatments [165].

The most commonly used instrument for measuring static blood pressure in clinical practice is the cuff sphygmomanometer [166]. But its method of automatically analyzing the arterial pulsation signal through cuff pressure fluctuations makes it difficult to achieve continuous dynamic measurements, and produces discomfort that interferes with daily life. For continuous blood pressure measurement, invasive fiber-optic pressure sensors are clinically implanted in the center of the artery, but this is often accompanied by pain and risk of infection and should not be used on a daily basis [167]. The emergence of wearable sleeveless blood pressure measurement devices has made it possible to continuously monitor blood pressure in daily life, and has therefore gained a lot of attention from researchers. Noninvasive methods that have been used to measure blood pressure include tonometry, ultrasound wall tracking, bioimpedance, and PPG [168]. Among them, the intraocular pressure measurement method shows unique advantages. Based on the principle of arterial deformation pressure conduction, this method has a streamlined sensor structure, low signal processing complexity (easier than the photoelectric signal processing of PPG), and breaks through the limitations of bioimpedance and ultrasound methods on the size of the device. Although it requires a high-sensitivity sensor and guarantees the stability of the interface, it has better overall wearability and provides a lightweight solution for ambulatory blood pressure monitoring. Li et al. reported a wearable system for continuous monitoring of blood pressure based on intraocular pressure measurement with an accuracy comparable to that of professional medical devices (Figure 6a). The components of this thin, soft, miniaturized system (TSMS) are two piezoelectric sensors encased in soft silicone (polydimethylsiloxane, PDMS, 145 kPa), an active pressure adaptation module that provides backpressure to improve interface stability, and a data processing module that determines the time difference in the pulse signals recorded by the two piezoelectric sensors in real time and sends the information to a graphical user interface (GUI). The TSMS is designed as a wearable wristband, with a thickness of only 4 mm and a weight of 50 g, making it lightweight, easy to wear, and soft to the touch. The sensor is positioned on the skin above the artery and accurately converts the local deformation caused by the pulse signal into an electrical signal. This signal is then calculated using tonometry in the data processing module and sent to the cell phone through the integrated BLE. The accuracy of the processed systolic and diastolic blood pressure data was −0.05 ± 4.61 mmHg and 0.11 ± 3.68 mmHg, respectively, surpassing the British Hypertension Society (BHS) Grade A standard. In clinical trials, researchers had 87 volunteers measure their blood pressure simultaneously using the TSMS system and commercial blood pressure monitors. The results showed that TSMS has extremely high accuracy, with over 98% of measurements having an error of less than 15 mmHg. Additionally, the device’s low power consumption ensures that, even after prolonged use, the operation temperature remains below 37 °C, preventing thermal discomfort and safeguarding the user from potential safety hazards [33].

The ratio of oxygen content to oxygen capacity in the blood, or the percentage of “oxygen-saturated hemoglobin” to “total hemoglobin,” is known as oxygen saturation (SO_2_) [169]. It can be used to evaluate the uptake and delivery of oxygen in tissues in order to determine the condition of tissue metabolism and microcirculation. Non-invasive oxygen saturation measurements in today’s clinics are typically performed using pulse oximetry. However, the output of this device is only arterial oxygen saturation (SaO_2_), and it is a rigid device that can be uncomfortable to wear for prolonged periods of time. Measurement of tissue oxygen saturation (rSO_2_), which is the local average oxygen saturation of hemoglobin in all red blood cells, often requires invasive sampling with fiber optic probes. Xin et al. present an optoelectronic skin biosensor capable of measuring several vital sign parameters such as rSO_2_, heart rate, arterial oxygen, and tissue perfusion within the mixed arteriovenous blood flow. The sensor can be used to analyze a variety of clinical symptoms such as lower limb ischemia, surgical vascular closure, and even skin ulceration (Figure 6b). The device consists of a biosensor part and a wireless measurement module connected by a flexible conductor. The biosensor part consists of three inorganic light-emitting diodes (LEDs) with light-emitting wavelengths of 660, 750, and 850 nm and two photodetectors (PDs), a serpentine honeycomb interconnecting wire, and a biocompatible package. During operation, three LEDs are alternately lit at a frequency of 500 Hz. At the same time, two PDs are responsible for monitoring the light reflected from the arteriovenous-rich dermal tissue at different locations by virtue of the device’s spatially resolved structure. By analyzing the information carried by six sets of reflected light of three wavelengths at two locations, blood information, such as rSO_2_, can be deduced. The device’s serpentine honeycomb interconnections and biocomponent (W1624; Tegaderm, 3M) package ensure that the device is tensile and flexible, and can be adsorbed to the skin for prolonged periods of time based solely on van der Waals forces. The device allows for easy and accurate diagnosis of peripheral arterial disease, as well as senseless postoperative monitoring. Researchers randomly selected 10 patients with severe, typical PAD and used the device to measure rSO_2_ at the site of the lesion. The final diagnosis accuracy rate was as high as 100%, including one case that the hospital had failed to diagnose correctly [23].

The use of wearable devices to monitor blood health marks a significant shift in modern medicine, moving away from the hospital-centered treatment model towards real-time monitoring and personalized care. However, wearable devices are still far from the clinical level in terms of accuracy due to their non-invasive design concept. Improving the measurement accuracy of wearable blood pressure and oximetry sensors is a goal that relevant researchers need to continue to strive for in the long-term future [170].

## 4. Summary and Prospects

In this review, we summarize recent research advancements in skin-inspired electronics for health monitoring in recent years, focusing on the selection of relevant advanced materials and communication techniques in terms of selection, preparation methods, and applications. The applications of skin-inspired electronics include temperature sensors, pulse sensors, blood pressure sensors, and oxygen sensors. Compared with the traditional health monitoring devices, the skin-inspired electronics are wearable, stretchable, and comfortable. By attaching to the skin surface or implanting into the human body, they provide diagnosis and early warning of various diseases, thus ensuring the physical health status of individuals. Combined with advanced wireless data transmission technology, they can achieve continuous, fast, real-time monitoring, and real-time data feedback and analysis.

Many achievements have been made in the application of skin-inspired electronics in the medical field. At the same time, commercial skin patches also began to appear on the market. In 2024, the smart skin system developed by Israel’s X-Rods Company passed FDA 510(k) certification, becoming the world’s first flexible, wearable patch to achieve multimodal, electrophysiological, synchronous monitoring. The ultra-thin, flexible patch can be concealed on the scalp, trunk, or limbs. It can seamlessly and synchronously collect electroencephalogram, ECG, electromyogram, and other biological potential data [171]. The intelligent vital signs monitoring patch developed by Catharina Hospital and Philips in the Netherlands is an innovative addition to surgical tumor nursing practice. During the pilot in the surgical ward, the patch automatically collected physiological data such as the patient’s heart rate, respiratory rate, body position, and activity status every five minutes. Following clinical equivalence verification, it replaced nurses’ manual measurements three times a day [172].

However, there are still many challenges in the development of the next generation of skin-inspired electronics for medical and health monitoring: (1) Real-time monitoring of SO_2_ in the human body mostly relies on optical sensors to obtain test metrics based on feedback and analysis of signals at different wavelengths. This imposes high demands on the quality of the sensors, especially skin-fitting sensors, and their resistance to interference from the environment and motion. (2) In addition to monitoring vital signs, various skin-like electronic devices used for health monitoring track parameters such as metabolites, disease marker concentrations, and human movement status. These devices are often exposed to body fluids and metabolites or worn on moving body parts, which presents challenges for maintaining functionality and accuracy. Therefore, selecting appropriate materials and encapsulating the devices to enhance their chemical stability in biological environments are feasible strategies. (3) The recording of various vital signs and chemical indicators of the human body generates a large amount of raw data accompanied by noise. Skin-like electronic devices need to use appropriate conversion circuits to reduce the noise from the actual application scene. At the same time, it is necessary to use big data technology to process the raw data and make early warnings based on established health indicator standards. (4) Traditional photolithography technology is not suitable for the preparation of large-area flexible devices. The current preparation process of high-performance skin-like electronic devices is more complex and costly, hindering their potential for mass production. Developing processes such as printed electronics, self-assembly, and template-assisted synthesis may be one of the solutions. (5) In addition to being unable to adapt to various environments and human movement patterns, wireless data transmission is still constrained by its use of the environment, service life, and distance. Moreover, it lacks all-weather senseless wear and accurate signal transmission/feedback. It is necessary to optimize the materials and structure of the wireless transmission module, upgrade the integration of transmission technology and energy harvesting, and optimize the real-time noise reduction algorithm.

By advancing material selection and molecular design, sensor system design and process optimization, as well as wireless data transmission system design and integration, it is anticipated that the challenges mentioned earlier will be overcome. Overall, wearable, stretchable, and comfortable skin-inspired electronics can monitor human health non-invasively, continuously, and in real time. They hold significant promise for the future of the health monitoring field, offering long-term benefits and potential.

## Figures and Tables

**Figure 4 biomimetics-10-00531-f004:**
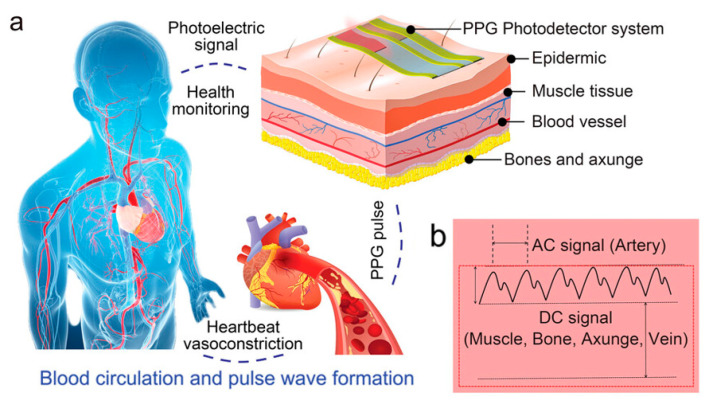
Mechanism of PPG signal, signal curve composition, and application scenarios. Reprinted with permission from Ref. [32] (copyright 2023 Wiley-VCH GmbH). (**a**) Diagram of the human blood circulation system and schematic diagram of PPG signal sensing on the body surface. (**b**) Structure of the basic curve of the PPG signal and the structure of the response of different parts of subcutaneous tissue.

**Figure 5 biomimetics-10-00531-f005:**
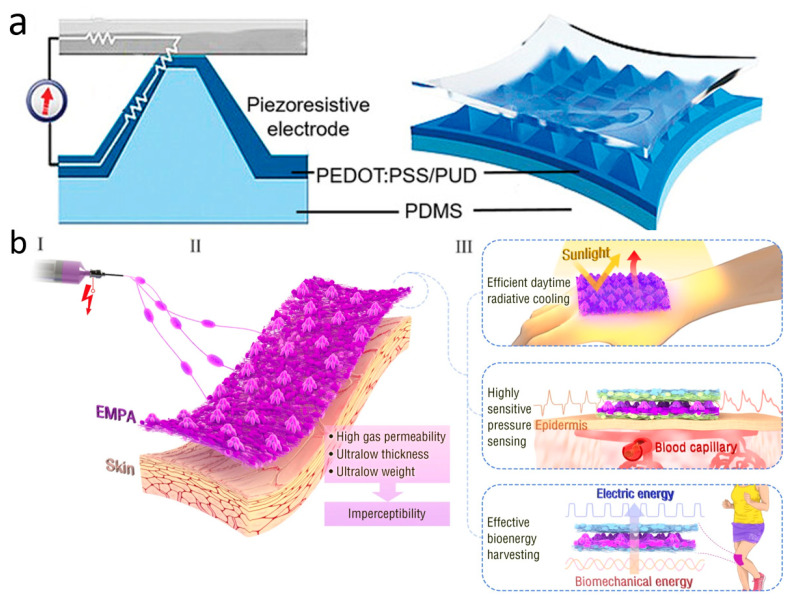
Wearable pulse pressure sensors. (**a**) Schematic diagram of wearable pulse sensor structure. Reprinted with permission from Ref. [160] (copyright 2022 Wiley-VCH GmbH). (**b**) Schematic illustration of the (I) fabrication, (II) structure, and (III) application of EMPAs. Reprinted with permission from Ref. [161] (copyright 2022 Jia-Han Zhang et al.).

**Figure 6 biomimetics-10-00531-f006:**
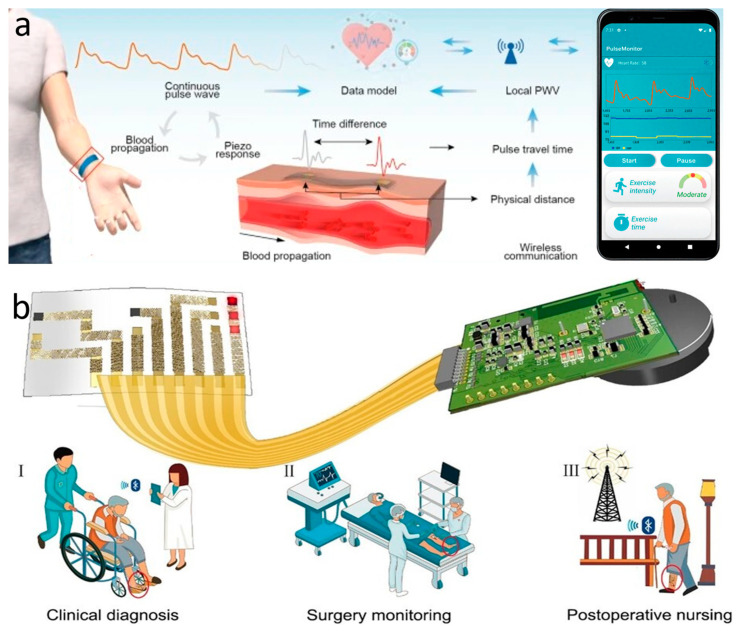
Blood pressure and oxygen sensors. (**a**) Wearable continuous blood pressure monitoring system based on intraocular pressure measurement. Reprinted with permission from Ref. [33] (copyright 2023 Jian Li et al.). (**b**) On-skin optoelectronic biosensor that measures tissue oxygen saturation. Expected to be attached to the red circle in the figure for (I) clinical diagnosis, (II) surgical monitoring and (III) postoperative care. Reprinted with permission from Ref. [23] (copyright 2022 Lijia Pan, Yi Shi, Xinran Wang et al.).

## Data Availability

No new data were created or analyzed in this study. Data sharing is not applicable to this article.

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
