# Peer review of "Skin-Inspired Healthcare Electronics"

_biomimetics, 2025, doi:10.3390/biomimetics10080531_

Round 1
Reviewer 1 Report
Comments and Suggestions for Authors
Conclusion: The manuscript requires major revision before acceptance.
1. The abstract should highlight the core innovation. The current description of the integration breakthrough of materials and wireless transmission technology is vague, and the key technical highlights need to be clarified.
2. The review of existing similar studies in the introduction is not comprehensive enough, and there is a lack of comparative analysis with the latest results, which makes it difficult to highlight the research value.
3. In the material chapter, the long-term stability of conductive hydrogels and the processing technology of liquid metals only mention problems, lacking specific experimental data support for improvement.
4. The wireless transmission part does not provide sufficient comparative analysis of the performance of NFC and BLE in medical scenarios (such as transmission stability and power consumption), and the discussion of application scenario adaptability is shallow.
5. In the application cases, clinical verification data for some sensors (such as blood oxygen sensors) is lacking, and the description of the measured effects in multiple scenarios is not detailed enough.
6. The feasibility analysis of the solutions proposed in the challenges and prospects section is insufficient, such as the specific path of large-scale production processes is unclear, and potential technical routes need to be supplemented.
Author Response
- Commend: The abstract should highlight the core innovation. The current description of the integration breakthrough of materials and wireless transmission technology is vague, and the key technical highlights need to be clarified.
- Reply: That makes a lot of sense. We modified abstract to reflect the highlights of this article.
- Commend: The review of existing similar studies in the introduction is not comprehensive enough, and there is a lack of comparative analysis with the latest results, which makes it difficult to highlight the research value.
- Reply: Thank you for the reminder. We surveyed recent relevant reviews and compared them with this article, highlighting the fact that this article provides a more systematic and comprehensive overview of the latest developments in e-skin in the healthcare field.
- Commend: In the material chapter, the long-term stability of conductive hydrogels and the processing technology of liquid metals only mention problems, lacking specific experimental data support for improvement.
- Reply: That's right. We added examples of using elastomer encapsulation and adding electrolytes to demonstrate progress in improving the long-term stability of hydrogels. Regarding liquid metals, we have listed examples of embedding liquid metal microchannels, photolithography after removing the oxide layer on the surface of liquid metals using high-concentration NaOH, and using 3D printing technology to solve processing problems.
- Commend: The wireless transmission part does not provide sufficient comparative analysis of the performance of NFC and BLE in medical scenarios (such as transmission stability and power consumption), and the discussion of application scenario adaptability is shallow.
- Reply: A comparison of NFC and BLE performance has been added to the last paragraph of Section 2.2. At the same time, we have modified the descriptions of some examples to better reflect their applicability to specific use cases.
- Commend: In the application cases, clinical verification data for some sensors (such as blood oxygen sensors) is lacking, and the description of the measured effects in multiple scenarios is not detailed enough.
- Reply: Thank you for pointing out. Clinical validation data have been added to several examples in the application, and the description of measurement effect has been supplemented.
- Commend: The feasibility analysis of the solutions proposed in the challenges and prospects section is insufficient, such as the specific path of large-scale production processes is unclear, and potential technical routes need to be supplemented.
- Reply: We have researched potential technical routes by consulting relevant literature, and finally supplemented the feasibility analysis of the solution.
Reviewer 2 Report
Comments and Suggestions for Authors
The manuscript titled "Skin-Inspired Healthcare Electronics" presents a comprehensive review of recent developments in flexible, skin-like electronic devices for vital sign monitoring. It begins with an in-depth discussion on material categories—nanomaterials, conductive hydrogels, and liquid metals—used in e-skin, followed by wireless signal transmission techniques like NFC and BLE. The review then highlights applications of these devices in monitoring temperature, pulse, blood pressure, and blood oxygen levels. It concludes by identifying current limitations, such as signal noise, material degradation, and challenges in mass production. The manuscript is well-organized and up-to-date, reflecting major technological and biomedical advances in wearable healthcare electronics.
Comment 1 : Although the review discusses signal acquisition and wireless transmission in detail, it lacks coverage of how the collected data are processed or analyzed—especially considering the growing role of machine learning in wearable health monitoring. A brief mention of signal preprocessing, noise filtering, or AI-based analysis pipelines would enhance the completeness of the paper.
Comment 2 : The section on material selection (Section 2.1) would benefit from a comparison table that summarizes properties of nanomaterials, hydrogels, and liquid metals (e.g., modulus, stretchability, conductivity, biocompatibility, durability). This would help readers more easily assess their relative advantages in different application scenarios.
Comment 3 : While many innovative materials and sensing strategies are discussed, the review does not reference any commercial products, clinical trials, or industry efforts. Including a few such examples (e.g., smart patches used in hospitals or FDA-cleared e-skin devices) would strengthen the practical relevance of the work.
Comment 4 :In the hydrogel section, classification by conduction mechanism (ionic, carbon-based, metal-based, polymer-based) is helpful but inconsistently detailed. The authors are encouraged to add specific examples with values for conductivity or Young’s modulus and summarize key differences in a diagram or figure.
Comment 5 : Figures 3–6 are well-designed; however, some are underexplained. For example, Figure 5b (page 16) describes the electrospun micro-pyramid array structure, but its role in signal enhancement and comfort is not sufficiently elaborated in the main text. This should be clarified.
Comment 6 : Some sentences could be more fluid or grammatically correct. For instance, on page 12, "the temperature sensors that are often chosen are thermistor temperature sensors..." could be revised for readability. A full proofread by a native English editor is recommended.
Author Response
- Commend: Although the review discusses signal acquisition and wireless transmission in detail, it lacks coverage of how the collected data are processed or analyzed—especially considering the growing role of machine learning in wearable health monitoring. A brief mention of signal preprocessing, noise filtering, or AI-based analysis pipelines would enhance the completeness of the paper.
- Reply: You're right. We have added section 2.3 to supplement the explanation of signal preprocessing and the role of AI-based analysis in wearable health monitoring.
- Commend: The section on material selection (Section 2.1) would benefit from a comparison table that summarizes properties of nanomaterials, hydrogels, and liquid metals (e.g., modulus, stretchability, conductivity, biocompatibility, durability). This would help readers more easily assess their relative advantages in different application scenarios.
- Reply: Indeed, such a table would be very helpful for readers to evaluate the relative advantages of different materials. However, nanomaterials and hydrogels each encompass a wide range of subtypes with highly variable characteristics. A rough comparison using a single table could lead to misleading generalizations. Therefore, I do not recommend inserting a comparison table here.
- Commend: While many innovative materials and sensing strategies are discussed, the review does not reference any commercial products, clinical trials, or industry efforts. Including a few such examples (e.g., smart patches used in hospitals or FDA-cleared e-skin devices) would strengthen the practical relevance of the work.
- Reply: Thank you for adding. We added two business examples in the summary and prospects section to enhance the practical relevance of the work.
- Commend: In the hydrogel section, classification by conduction mechanism (ionic, carbon-based, metal-based, polymer-based) is helpful but inconsistently detailed. The authors are encouraged to add specific examples with values for conductivity or Young’s modulus and summarize key differences in a diagram or figure.
- Reply: Sorry, the content of this chapter is not clear enough, causing your misunderstanding. We are only going to introduce conductive polymer hydrogels. Through different processes, conductive polymer hydrogels can show different advantages in conductivity and extensibility. Therefore, longitudinal comparison is of little significance. In contrast, we have adjusted the language context of this chapter. The actual data are added in the place where the conductivity is prominent.
- Commend: Figures 3–6 are well-designed; however, some are underexplained. For example, Figure 5b (page 16) describes the electrospun micro-pyramid array structure, but its role in signal enhancement and comfort is not sufficiently elaborated in the main text. This should be clarified.
- Reply: Thank you for your praise and correction. We have added relevant descriptions to the text to ensure that the pictures are explained.
- Commend: Some sentences could be more fluid or grammatically correct. For instance, on page 12, "the temperature sensors that are often chosen are thermistor temperature sensors..." could be revised for readability. A full proofread by a native English editor is recommended.
- Reply: Thank you for your feedback. We have revised several sentences throughout the text to improve readability.